# PROBMEDTOD: A BAYESIAN NETWORK GUIDED TASK-ORIENTED DIALOGUE SYSTEM FOR PATIENT HISTORY TAKING

## ABSTRACT

Task-oriented dialogue (TOD) systems for patient history-taking improve clinical workflow efficiency by collecting key diagnostic information. Most data-driven approaches for this rely on large language models (LLMs) and mimic fast, intuitive System 1 thinking. In contrast, clinicians typically reason about potential diagnoses and use that to guide the dialog.

To bridge this gap, we propose ProbMedTOD, a TOD system that combines the conversational abilities of LLMs with the probabilistic reasoning of a disease-symptom Bayesian Network (BayesNet). At each turn, ProbMedTOD extracts information from patient utterances, updates its diagnostic hypothesis over a set of potential principal diagnoses via Bayesian inference, and generates the next question using a supervised policy LLM trained on dialogue data. The BayesNet structure is programmatically constructed from clinical documents, while its parameters are inferred automatically via self-consistent prompting of an LLM, removing the need for expert-labeled data.

We develop a patient simulator that uses patient profiles informed by the dialogue context and engages in realistic end-to-end interactions with the system, enabling evaluation of dialogue-level success. ProbMedTOD significantly outperforms LLM and retrieval-based baseline in next-question prediction and dialogue-level success, obtaining 20 pt MRR improvement in simulation experiments.

Patient history taking is a critical step in clinical diagnosis (Engel & Morgan, 1973; Hampton et al., 1975; Boyd & Heritage, 2006). It involves a strategic conversation to narrow down on a principal diagnosis, condition chiefly responsible for the patient's visit (CMS, 2024). This process is inherently uncertain as symptoms may be ambiguous or incomplete, requiring clinicians to iteratively update their diagnostic hypotheses. For example, a persistent cough could suggest a common cold, tuberculosis, or even lung cancer. To refine diagnoses, a doctor must consider plausible conditions by their likelihoods and select future inquiries. Figure 1 illustrates this process where the doctor decides the next inquiry (Fatigue) to refine diagnostic belief (Tuberculosis).

## 1 INTRODUCTION

A TOD system for patient history-taking must imitate this process through multi-turn interactions. While recent systems based on LLMs have achieved impressive fluency (Dou et al., 2023; Xu et al., 2024; He et al., 2024) and can ground themselves in clinical knowledge bases (Xu et al., 2023; Sree et al., 2024; Varshney et al., 2025a), they generally do so by learning patterns in the training data. Such an approach is reflective of System 1 thinking. They lack explicit reasoning mechanisms to manage uncertainty robustly. They also do not inherently model how evidence accumulates over time, how beliefs over principal diagnoses should probabilistically change as new information arrives, or how to select informative inquiries to efficiently improve diagnosis in a principled way.

To address this critical gap, we propose ProbMedTOD, a medical dialogue system that treats history taking as decision-making informed by probabilistic inference, aligning with the System 2 thinking of medical practitioners. At each dialogue turn, ProbMedTOD employs a pipeline of specialized modules. At each turn, a Natural Language Understanding (NLU) component extracts structured

**Context:** Patient reports a persistent **cough** for approximately three weeks. Initially, the cough presented with white to yellowish sputum (**colored/excess sputum**). Over the past week, the patient has developed **hemoptysis**, estimating the amount to be about half a teaspoon. The patient denies experiencing **dyspnea, chest pain, wheezing, rhinorrhea, or pharyngitis**. The patient does report **chest discomfort** associated with the cough.

**Gold Response:** How has your energy been? Any any fatigue?

**Llama3 8B:** Do you have a headache?

**Llama3 8B + RAG:** Have lost you sense of smell or taste?
**Retrieved Documents:**
1. **Pertusis** causes coughing, loss of consciousness, whooping cough,....
2. **Asthma** causes chest discomfort, colored/excess sputum, dyspnea, coughing,....
3. **Bronchitis** causes chest discomfort, coughing, fatigue,....

**ProbMedTOD:** Do you have chills or fever?
**Bayesian Inference:**
1. **Tuberculosis (89%)** causes coughing (82.7%), hemoptysis (36.3%), fever (76.5%),...
2. **Bronchitis (5%)** causes chest discomfort (53.4%), coughing (86.9%), fatigue (49.2%),....
3. **Pneumonia (5%)** causes coughing (90.4%), hemoptysis (90.4%), chills (67.9%)

Figure 1: Doctor iteratively refines a diagnostic hypothesis (Tuberculosis) by asking targeted questions. ProbMedTOD aims to replicate this process using a disease-symptom BayesNet.

symptoms from the patient's utterance to update a distribution over possible diagnoses via Bayesian inference on a disease-symptom Bayesian Network (BayesNet). A supervised dialogue policy then selects the next inquiry for a Natural Language Generation (NLG) component to verbalize. After the dialogue, ProbMedTOD uses its final posterior distribution to output the most likely principal diagnosis.

Bayesian networks are well suited to medical diagnosis, capturing the probabilistic dependencies between diseases and symptoms and allowing principled belief updates as new evidence arrives (Polotskaya et al., 2024). They also provide a transparent reasoning process, which is crucial for clinical trust. While the network structure can often be derived from domain knowledge, estimating its parameters usually requires large-scale patient data (which has privacy implications). ProbMedTOD addresses this challenge by leveraging an LLM as a novel source for estimating key parameters, disease priors and symptom likelihoods, through structured prompting and self-consistency sampling, enabling the construction of a reliable probabilistic model without extensive datasets.

Since the goal of a history-taking system is to conduct effective patient interviews and produce an accurate principal diagnosis, conventional dialogue metrics (e.g., BLEU, Intent F1) are insufficient. They evaluate only turn-level conversational quality and may penalize valid questions asked later. To address this, we introduce a notion of task success based on patient cases sourced from MediTOD (Saley et al., 2024) and MIMIC-IV (Johnson et al., 2023) datasets, and develop a user simulator that uses these profiles for high quality conversations. This allows realistic simulations to measure dialogue-level performance. In this environment, ProbMedTOD consistently outperforms strong baselines on principal diagnosis accuracy, demonstrating the value of explicit probabilistic modeling and learned dialogue strategies in medical dialogue systems.

In summary, our main contributions are as follows. (1) We present ProbMedTOD, a hybrid TOD system that integrates an NLU component, BayesNet-based probabilistic inference, and a learned dialogue policy for robust patient history taking under uncertainty. (2) We develop a novel and efficient method for automatically estimating parameters of a disease-symptom BayesNet directly from an LLM using structured self-consistency prompting, eliminating the need for manual knowledge engineering or large structured medical datasets. (3) We design a robust simulation-based evaluation framework with patient profiles grounded in MediTOD and extended with MIMIC-IV, providing realistic and scalable benchmarks for diagnostic reasoning. (4) We perform evaluation comparing our model with two types of baselines (LLMs without BayesNet and LLMs with retrieval-augmented generation) on next utterance and principal diagnosis prediction tasks to find that ProbMedTOD outperforms all baselines on both tasks.

## 2 PROBLEM DEFINITION & SOLUTION APPROACHES

We now formally define the task of a TOD system for patient history taking. Let $H_t = \{u_1, s_1, u_2, s_2, \ldots, u_t\}$ denote dialogue history up to turn $t$, where $u_i$ and $s_i$ represent the patient and system (doctor) utterances, respectively. The objective is to generate the next system response $s_t$ that strategically gathers information to produce an accurate principal diagnosis (CMS, 2024) at the end of the dialogue.

A standard design decomposes TOD into three supervised modules: Natural Language Understanding (NLU), Policy (POL), and Natural Language Generation (NLG) (Young et al., 2013; Hosseini-Asl et al., 2020; Lin et al., 2020). The NLU (Natural Language Understanding) module extracts semantic information from the patient's utterance $u_t$, such as intents (e.g., inform, inquire) and slot-value pairs (e.g., symptom–fever, past medical history–diabetes). This information accumulated across dialogue turns into a dialogue state $\text{dst}_t$. Based on the dialogue history $H_t$ and the current state $\text{dst}_t$, the POL (Policy) module selects the next action $a_t$ (e.g., inquire about a new symptom). Finally, the NLG (Natural Language Generation) module converts $a_t$ into a natural language response $s_t$ (e.g., Do you have a fever?).

Recent approaches model these components by fine-tuning LLMs on a dataset $\mathcal{D} = \{(H_t, \text{dst}_t, a_t, s_t)\}$, consisting of annotated doctor-patient dialogue turns. In this work, we focus on the POL module, which is critical to the TOD system's performance, as it determines the next question to ask and thus defines the flow of the interaction.

To support diagnostic reasoning, an external corpus consisting of documents describing diseases $\{D_1, D_2, \ldots, D_N\}$ and their associated symptoms is often available. Retrieval-augmented generation (RAG) (Lewis et al., 2020; Sree et al., 2024; Xu et al., 2024) methods select top-$k$ documents from the corpus using $H_t$, and condition the policy on these retrieved documents to generate the next action.

However, such approaches typically treat documents as flat contexts and do not explicitly model the probabilistic relationships between diseases and symptoms. This limits their ability to reason under uncertainty or update diagnostic beliefs as new evidence is collected. An effective TOD system for history-taking must not only retrieve relevant knowledge but also reason about the evolving likelihoods of candidate conditions to select maximally informative next questions.

ProbMedTOD is composed of two main components: (1) a probabilistic reasoning module based on a disease-symptom Bayesian Network (BayesNet), and (2) a policy model that selects the next system action using both the dialogue history and the current diagnostic belief. At each dialogue turn, ProbMedTOD uses symptoms specified in $\text{dst}_t$ to form a diagnostic hypothesis $BN_t$ via Bayesian inference. This diagnostic hypothesis, $BN_t$, is then passed to a policy language model that generates the next doctor action. Figure 2 illustrates the overall design of ProbMedTOD.

### 2.1 BAYESNET

**Structure and Parameterization**  To capture probabilistic relationships between diseases and symptoms while avoiding the intractability of modeling a full joint distribution, we employ a Bayesian Network (BayesNet) with a Noisy-OR parameterization (Jaakkola & Jordan, 1999). The network takes the form of a bipartite graph as shown in figure 3a. It consists of binary disease nodes $\{D_1, D_2, \ldots, D_N\}$ and binary symptom nodes $\{S_1, S_2, \ldots, S_M\}$, a structure that naturally encodes the causal flow from diseases to their symptoms. Such a representation is well suited to medical reasoning, as it combines interpretability with efficient inference in high-dimensional settings.

The model have three types of parameters: disease priors $p_j = P(D_j = 1)$, representing the marginal probability of each disease; causal probabilities $q_{ji}$, which specify the likelihood that an active disease $D_j$ independently triggers symptom $S_i$; and leak probabilities $\lambda_i$, which account for unobserved or unknown triggers of $S_i$. Under the Noisy-OR assumption, if $D = (D_1, \ldots, D_N)$ is a binary vector of disease states ($D_j \in \{0, 1\}$) and $\text{pa}(S_i)$ denotes the set of (parent) disease nodes causally connected to symptom $S_i$, then

$$P(S_i = 1 \mid D) = 1 - (1 - \lambda_i) \prod_{j \in \text{pa}(S_i)} (1 - q_{ji})^{D_j}. \tag{1}$$

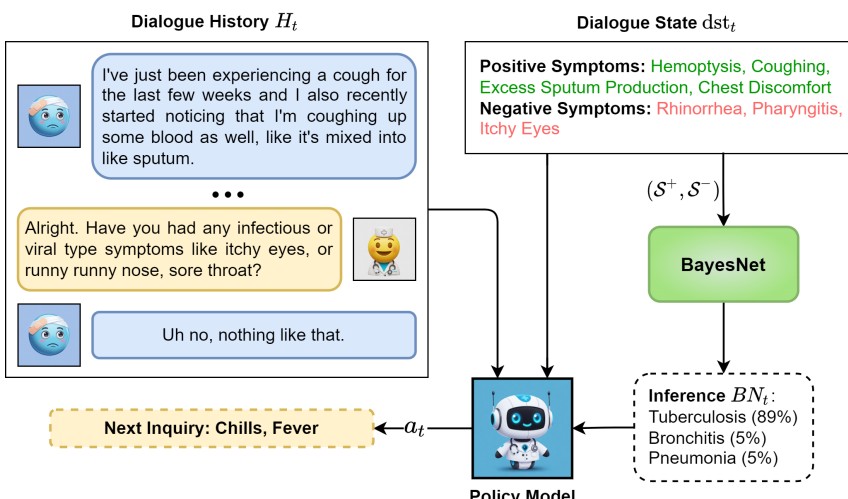

Figure 2: ProbMedTOD architecture. Dialogue history and extracted symptoms are used to infer disease posteriors via a BayesNet. The top diseases are fed into the policy model along with the dialogue state to generate the next inquiry.

We construct the network from publicly available clinical corpus by programmatically extracting disease–symptom edges, followed by manual validation and correction as needed. The resulting network contains 34 diseases, 123 symptoms, and 376 directed edges.

## 3 PROBMEDTOD SYSTEM

**Parameter Estimation** LLMs have recently shown strong performance across a range of clinical tasks, including medical question-answering and diagnostic reasoning (Singhal et al., 2025; Nori et al., 2023; Pal & Sankarasubbu, 2024). These capabilities suggest that LLMs encode structured medical knowledge, implicitly capturing associations between diseases and symptoms. We leverage this insight to prompt an LLM to directly estimate disease priors $p_j$ and symptom conditional likelihoods $q_{ji}$, eliminating the need for patient-level data or expert-curated annotations.

The prompt templates used for estimation are provided in Appendix B. Each prompt asks the LLM to return a numerical value corresponding to either the disease prevalence or the likelihood of a symptom given a disease. To improve the stability of our estimation, we apply self-consistency prompting (Wang et al., 2023), sampling 50 completions for each query. We use a temperature of 1.0 to maximize the diversity among the completions. The final probability for each parameter is computed as the mean across samples.

**Inference** At each dialogue turn $t$, ProbMedTOD extracts the set of positive symptoms $\mathcal{S}^+$ and negative symptoms $\mathcal{S}^-$ from the dialogue state $\text{dst}_t$, and computes the posterior probability of each disease given this evidence: $P(D_j = 1 \mid \mathcal{S}^+, \mathcal{S}^-)$.

Direct computation of this posterior is generally intractable due to the exponential complexity of marginalizing over all possible disease combinations. Since our primary objective is to identify the principal diagnosis, we adopt a single-disease generative model – we assume one disease to be chiefly responsible for the patient's visit (Guan & Baral, 2021). This assumption greatly simplifies inference, but is not a limitation of our framework as BayesNets can naturally be extended to support multiple concurrent diagnoses. Under the single-disease assumption, the posterior simplifies to:

$$P(D_j = 1 \mid \mathcal{S}^+, \mathcal{S}^-) \propto \ P(D_j = 1, D_{j^-} = 0)$$
$$\times \prod_{i \in \mathcal{S}^+} P(S_i = 1 \mid D_j = 1, D_{j^-} = 0) \times \prod_{i \in \mathcal{S}^-} P(S_i = 0 \mid D_j = 1, D_{j^-} = 0) \quad (2)$$

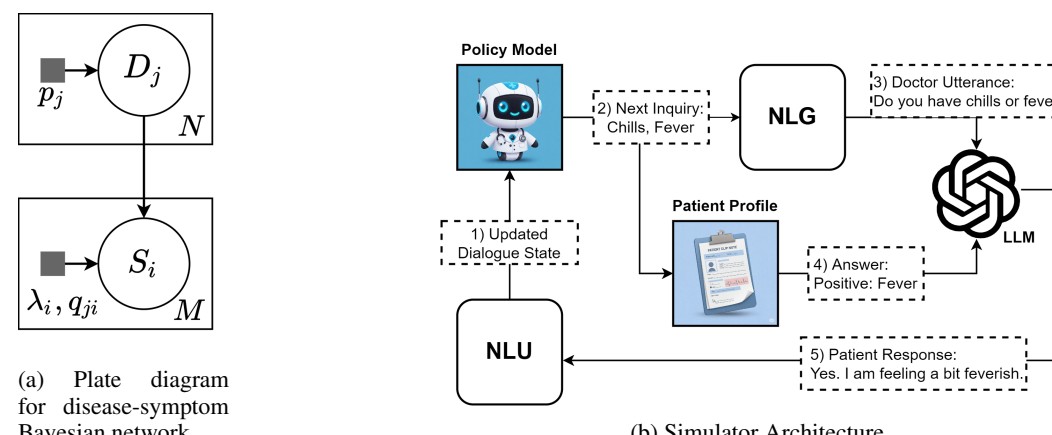

(a) Plate diagram for disease-symptom Bayesian network

(b) Simulator Architecture

Figure 3: The BayesNet structure and the simulator design

where $D_{j^-} = 0$ denotes that all other diseases are inactive. The symptom likelihoods $P(S_i \mid D_j = 1, D_{j^-} = 0)$ are computed using the Noisy-OR model as in equation 2.1. This simplification allows posterior inference to be performed efficiently in $O(N * |\mathcal{S}^- \cup \mathcal{S}^-|)$ time. A detailed derivation is provided in Appendix C.

## 3.1 POLICY MODEL

The goal of the policy model in ProbMedTOD is to predict the next doctor action $a_t$ based on the dialogue context and the current diagnostic hypothesis. At each turn, ProbMedTOD computes the posterior distribution over diseases using the BayesNet, as described earlier. It then selects the top-$k$ most probable diseases as a diagnostic hypothesis set $BN_t$. For each disease in this set, ProbMedTOD constructs a short note that includes the disease name, its posterior probability, and a brief description from the corpus of diseases. This note is then combined with the dialogue history $H_t$ and the dialogue state $dst_t$, and passed as input to the policy model. The policy model is a language model that outputs the next action $a_t$.

Including numerical posterior probabilities in the input allows the policy model to interpret how likely each disease is, rather than treating all diagnoses equally. This helps it focus on the most relevant conditions, ask more informative questions, and make better decisions under uncertainty. We experiment using LLaMA3 8B (Grattafiori et al., 2024) and Qwen3 8B (Team, 2025) as our policy models and fine-tune then on a dataset of annotated medical dialogues, where each dialogue turn is labeled with the corresponding doctor action.

## 4 PATIENT SIMULATOR

Evaluating medical TOD systems is essential to ensure they support accurate diagnosis. However, existing metrics typically compare predicted actions to a fixed "gold standard" dialogue, failing to account for the variability in valid clinical reasoning. For example, a tuberculosis patient may present with fever and chills; asking about either is valid, yet current metrics may penalize it.

We propose a patient simulator for more robust evaluation. The simulator engages a TOD system in multi-turn interactions and assesses whether it can gather relevant information to reach the correct diagnosis. To support diverse reasoning paths, we construct complete and clinically accurate patient profiles, specifying all active symptoms, medical history, habits, and other relevant details. An LLM then generates fluent, faithful responses grounded in these profiles. The overall working of our simulator is shown in Figure 3b.

**Patient Profile Construction** We begin by constructing patient profiles from the MediTOD dataset (Saley et al., 2024), which contains 213 doctor–patient dialogues. A team of professional annotators with backgrounds in biology and pharmacology first labeled a subset of these dialogues

following the annotation guidelines of Saley et al. (2024). Their annotations were reviewed by a medical doctor, and the highest-performing annotators were then tasked with assigning principal diagnoses for the remaining cases. These diagnoses serve as anchors for building the knowledge corpus and BayesNet used in our experiments.

To generate patient profiles for the simulator, we draw from the MediTOD test dialogues. For each case, we use the known principal diagnosis and enrich it with additional, clinically plausible symptoms. For example, if a patient is diagnosed with sinusitis and reports headache, we augment the profile with sinus pressure as a co-occurring symptom. To ensure consistency and medical validity, we intermittently use the Gemini-2.5-Pro model to support profile generation.

**MIMIC-IV Patient Profiles**    While the MediTOD test set provides high-quality dialogues, it contains only 18 cases, limiting scale of our evaluations. In response, we turn to the publicly available MIMIC-IV Clinical Note dataset (Johnson et al., 2023), a large, de-identified repository of hospital records. From this corpus, we filter for patients' first hospital visits and select cases whose ICD-9/10 diagnosis codes align with the diseases present in MediTOD. For each disease, we sample up to three distinct patient cases to balance diversity with manageable evaluation cycles. Patient profiles are then constructed by bootstrapping key information from the clinical notes, such as the chief complaint, history of present illness, and past medical history.

**Simulator Workflow**    Figure 3b shows our simulator in action. At each turn, the policy model receives the current dialogue state from a trained NLU module and generates the next action such as a symptom inquiry. This action is passed to the trained NLG module to generate the corresponding doctor utterance (e.g., "Do you have chills or fever?").

The same system-generated action is used by the simulator to query the patient profile for relevant symptom values (e.g., Positive: Fever). Then, an LLM produces a fluent, natural-sounding patient response that remains faithful to the profile (e.g., "Yes, I've been feeling feverish lately."). This response is parsed by the NLU to update the dialogue state, and the interaction continues.

We use tailored prompts based on the type of doctor action to ensure medical coherence and profile consistency throughout the simulation. Appendix E provides further details.

**Evaluation**    After each simulated patient dialogue, the TOD system analyzes the collected clinical information to produce a ranked list of potential principal diagnoses. We train our POL model over MediTOD train set to predict a principal diagnosis when prompted. To generate a ranked list, we use Beam search with width five to collect various diagnoses.

For comparison, we benchmark our system against a Retrieval-Augmented Generation (RAG) method, which ranks diseases based on retrieved medical knowledge. We then evaluate the accuracy of the rankings against the correct principal diagnosis using standard retrieval metrics like Mean Reciprocal Rank (MRR) and Hit@K. These metrics measure how prominently the correct principal diagnosis appears in the output, emphasizing alignment with effective clinical reasoning. Unlike conventional metrics that track surface-level actions, our approach rewards systems for collecting relevant evidence through diverse diagnostic paths.

## 5 EXPERIMENTAL SETUP

**Dataset**    We use official train/valid/test splits of MediTOD dataset (Saley et al., 2024) for our experiments. MediTOD contains English doctor-patient dialogues from pulmonary specialty, where each utterance has comprehensive action annotations. As discussed in the last section, we assign each dialogue a gold principal diagnosis label, a condition chiefly responsible for the patient's visit. Details are given in Appendix A.

**Evaluation Metrics**    We conduct a multi-faceted evaluation to validate our approach. Following Saley et al. (2024), we perform turn-level evaluations on the provided test set and report *Medical F1* and *Precision@K* metrics. Medical F1 compares predicted actions with gold annotations at each turn. Precision@K matches them with any of the next $K$ gold actions. While Medical F1 is strict, Precision@K allows for flexibility in dialogue paths. To evaluate diagnostic utility, we simulate doctor–patient dialogues and report *MRR* and *Hit@K*.

| Model | Medical | Precision@ | | | |
|---|---|---|---|---|---|
| | F1 | 1 | 4 | 8 | Inf |
| PPTOD (base)* | 0.210 | 0.153 | 0.284 | 0.326 | 0.359 |
| Flan-T5 (base)* | 0.203 | 0.115 | 0.204 | 0.238 | 0.265 |
| BioGPT* | 0.185 | 0.135 | 0.251 | 0.300 | 0.348 |
| OpenBioLLM 8B* | 0.217 | 0.161 | 0.295 | 0.338 | 0.374 |
| Llama3 8B* | 0.239 | 0.188 | 0.318 | 0.382 | 0.416 |
| RAG | 0.257 | 0.195 | 0.327 | 0.390 | 0.423 |
| ProbMedTOD (Llama3 8B) | 0.262 | 0.209 | 0.357 | 0.408 | 0.442 |
| ProbMedTOD (Qwen3 8B) | **0.273** | **0.219** | **0.364** | **0.415** | **0.453** |

Table 1: ProbMedTOD performance on MediTOD Policy Learning Task. The highest and second-highest scores are bolded and underlined, respectively. * - results taken from Saley et al. (2024).

**Baselines**  We compare our approach against several supervised policy models, including PPTOD (Su et al., 2022), Flan-T5 (Longpre et al., 2023), OpenBioLLM (Ankit Pal, 2024), BioGPT (Luo et al., 2022), and LLaMA3 8B (Grattafiori et al., 2024).

Our primary baseline, however, is a Retrieval-Augmented Generation (RAG) system, which leverages external medical knowledge. Specifically, we implement RAG using a BGE-M3 (Chen et al., 2024) pre-trained retriever to perform maximum inner product search over the corpus of disease documents. The query consists of the dialogue history concatenated with the current dialogue state. The top-5 retrieved documents are then passed, along with the original query, as input to a LLaMA3 8B policy model, which is fine-tuned on MediTOD train set.

**Implementation Details**  We train all our models with LLaMA3 8B (Grattafiori et al., 2024) and Qwen3 8B (Team, 2025) as the backbone LLMs. For efficient training, we adopt the Unsloth framework (Daniel Han & team, 2023). All models are fine-tuned using LoRA (Hu et al.) with hyperparameters: rank $r = 32$, LoRA $\alpha = 128$, a learning rate of $5 \times 10^{-5}$ and a batch size of 16. Training is conducted for 10 epochs on a single A100 GPU, with each policy model taking about twelve hours to train. For RAG and ProbMedTOD, we use the top five diseases to form the diagnostic hypothesis for the policy model. For all models, we use greedy decoding to generate outputs.

Parameter estimation is performed using Gemma 2 27B IT (Team et al., 2024), chosen for its public availability, cost-efficiency, and single-GPU inference with vLLM (Kwon et al., 2023). While performance is satisfactory, proprietary models like GPT-4o may yield better estimates at higher computational costs. We set all leak probabilities $\lambda_i$ in the BayesNet to a small constant $10^{-4}$ decided based on the performance on validation set.

**Simulator Settings**  We evaluate our approach using a patient simulator, training its NLU and NLG on the MediTOD train set. Patient responses are generated with the MedGemma 27B model (Sellergren et al., 2025) at temperature 0.0 for reproducibility.

# 6  RESULTS

**Turn-level Results**  Table 1 summarizes policy generation performance across different models on the MediTOD test set. Our proposed models, ProbMedTOD (LLama3 8B) and ProbMedTOD (Qwen3 8B), consistently outperform all baselines on both Medical F1 and Precision@K metrics.

RAG remains the strongest baseline due to access to structured external knowledge. However, ProbMedTOD models achieve substantial improvements, with ProbMedTOD (Qwen3 8B) showing a 16 points gain in Medical F1 over RAG. The advantage of ProbMedTOD is particularly pronounced across Precision@K, with the gap consistent at higher values of $K$, highlighting that our probabilistic approach generates inquiries that are not only immediately relevant but also more informative and diagnostic for future dialogue turns.

Overall, the results indicate that incorporating probabilistic reasoning improves policy learning, leading to more informative and diagnostic actions during the dialogue.

| Model | MediTOD | | | MIMIC-IV | | |
|---|---|---|---|---|---|---|
| | MRR | Hit@1 | Hit@5 | MRR | Hit@1 | Hit@5 |
| RAG | 0.303 | 0.111 | 0.556 | 0.261 | 0.119 | 0.369 |
| ProbMedTOD (Llama3 8B) | 0.491 | **0.444** | 0.556 | 0.281 | **0.226** | 0.357 |
| ProbMedTOD (Qwen3 8B) | **0.500** | **0.444** | **0.611** | **0.303** | 0.214 | **0.452** |

Table 2: Simulation Results: ProbMedTOD achieves higher MRR and Hit@K compared to RAG.

| Model | MediTOD | | | | MIMIC-IV | |
|---|---|---|---|---|---|---|
| | Med F1 | Precision@Inf | MRR | Hit@5 | MRR | Hit@5 |
| ProbMedTOD (Qwen3 8B) | 0.219 | 0.453 | 0.500 | 0.611 | 0.303 | 0.452 |
| w/o BayesNet | 0.197 | 0.436 | 0.303 | 0.389 | 0.230 | 0.345 |

Table 3: Ablation Results

**Simulation Results**   To further assess the diagnostic effectiveness of the policy models, we evaluate them in our simulation framework. In this setting, the end goal is to accurately identify the patient's principal diagnosis after a full dialogue interaction with the simulator. We report MRR and Hit@K as evaluation metrics.

Table 2 shows the comparative performance of ProbMedTOD and the RAG baseline. Both ProbMedTOD variants outperform RAG across most metrics. These improvements suggest that integrating probabilistic reasoning is beneficial for this task. The Qwen3 8B variant, in particular, emerges as the superior model. On the MediTOD dataset, it achieves a MRR of $0.500$ and a Hit@5 of $0.611$. This trend extends to the MIMIC-IV dataset, where the Qwen3 model again leads with an MRR of $0.303$ and a Hit@5 of $0.452$. Dominance of Qwen3 8B over Llama3 8B indicates that Qwen3 adapts to diagnostic hypothesis better. Finally, all models exhibit weaker performance on the MIMIC-IV dataset compared to MediTOD, indicating a clear need for further improvements to enhance generalization across different clinical datasets.

**Ablation Study**   To access the contribution of our probabilistic reasoning module, we perform an ablation study using our best-performing model, ProbMedTOD (Qwen3 8B). For this study, we remove the BayesNet, creating a variant that relies solely on an LLM-only policy model to interact with the patient and perform the principal diagnosis at the end.

The results in table 3 clearly show that removing the BayesNet leads to a significant degradation in performance across all metrics on both datasets. On MediTOD, for example, the MRR drops by 19 points, and the Hit@5 score falls sharply by 22 points. A similar decline is observed on the MIMIC-IV dataset. This sharp performance drop shows that BayesNet's explicit probabilistic reasoning is integral to the model's diagnostic accuracy.

**Parameter Validation**   Although our results indicate LLM-estimated parameters are sensible, their direct assessment is challenging. To provide more evidence, we divide Bayesian network edges into primary (273) and other (103) based on disease corpus. Our strategy yields an average likelihood estimate across edges of 0.59 for primary vs. 0.34 for other (difference = 0.250), compared to a random baseline of 0.51 vs. 0.53 (difference = -0.02). For example, for whooping cough, the primary symptom coughing receives a higher estimate (0.948) than the secondary symptom fever (0.278), illustrating that our parameters meaningfully capture symptom relevance.

## 7   QUALITATIVE ANALYSIS

Appendix D) provides a simulated doctor-patient dialogue snippet with ProbMedTOD (Qwen3-8B) as the doctor. We find the the mode demonstrates a structured and clinically attentive approach. It systematically gathers key details about the cough, including onset, duration, type, and presence of mucus. Notably, it asks high-yield questions, such as the presence of blood (hemoptysis) and nocturnal shortness of breath (paroxysmal nocturnal dyspnea, PND), providing crucial information

to rule out serious conditions like tuberculosis, lung cancer, or heart failure. This illustrates model's attempt to refine the differential diagnosis aligned with System 2 thinking. Overall, the dialogue exhibits a logical flow, focusing on critical clinical indicators while attending to the patient's fatigue and weakness, reflecting a contextually appropriate and diagnostic-oriented consultation. However, we find as the conversation progress, the model asks non-informative questions.

## 8    RELATED WORK

**Medical Dialogue Systems**    With the rise of LLMs, their application in medical tasks has expanded rapidly, showing strong performance in medical QA and diagnosis benchmarks (Chen et al., 2023; Savage et al., 2024; Pal & Sankarasubbu, 2024; Tu et al., 2024; Singhal et al., 2025). Recent work has focused on adapting LLMs to medical dialogue settings. Li et al. (2023) fine-tuned LLaMA on doctor-patient conversations to build a QA system. Other approaches, such as Xu et al. (2024) and He et al. (2024), use chain-of-thought prompting (Wei et al., 2022) to capture clinical reasoning, distilling this into smaller models. Several systems (Sree et al., 2024; Varshney et al., 2025b) also integrate external medical knowledge through retrieval-augmented generation (RAG) (Lewis et al., 2020); for example, Xu et al. (2024) uses RAG to infer likely diagnoses from dialogue context.

However, none of these systems explicitly model diagnostic uncertainty. They rely on medical relationships learned during training, without probabilistic reasoning or belief updates. In contrast, ProbMedTOD maintains a diagnostic hypothesis using a Bayesian network, enabling principled updates and structured decision-making under uncertainty.

**BayesNets in Medical Diagnosis**    BayesNets have a long history in medical diagnosis, offering a principled way to model uncertainty and reason under incomplete information. Seminal efforts include the probabilistic reformulation of the INTERNIST-1/QMR system (Shwe et al., 1991) and the use of variational inference for scalable reasoning in the QMR-DT network (Jaakkola & Jordan, 1999). More recent work highlights the continued relevance of BayesNets in healthcare (Polotskaya et al., 2024).

Two closely related works are those by Guan & Baral (2021) and Pavez & Allende (2024). Guan & Baral (2021) use a Bayesian framework to select the next symptom to inquire based on expected information gain, aiming to reduce diagnostic uncertainty. However, their system operates on structured inputs and lacks conversational capabilities.

Pavez & Allende (2024) combine a BERT-based classifier with a Bayesian Network for mental health diagnosis, using BERT to generate a prior over diseases that is refined via probabilistic inference. While both approaches share our focus on uncertainty-aware reasoning, they do not leverage natural language dialogue or learn from conversational data. In contrast, ProbMedTOD integrates Bayesian inference with the fluent interaction abilities of LLMs, enabling data-driven, interpretable, and conversational medical history-taking.

## 9    CONCLUSION

In this work, we introduced ProbMedTOD, a hybrid task-oriented dialogue system for medical history taking that integrates LLM–based policy learning with explicit probabilistic reasoning via a disease–symptom Bayesian Network. ProbMedTOD employs a novel method for automatically estimating BayesNet parameters from an LLM through structured self-consistency prompting, removing the need for large-scale patient datasets and expensive annotations. We also create a patient simulation framework for computing dialogue level metrics; now, models do not get unfairly penalized if they ask the same symptom at a later utterance. Through comprehensive experiments on the MediTOD and MIMIC-IV datasets, ProbMedTOD outperforms strong LLM and RAG baselines on both turn-level policy metrics (Medical F1 and Precision@K) and diagnostic metrics within our simulation framework (MRR and Hit@K). Ablation study highlights the critical role of the BayesNet in improving both inquiry quality and final diagnosis accuracy. We will release both the code and dataset for further research.

## REPRODUCIBILITY STATEMENT

To ensure reproducibility, we fixed the random seed to 44 across all experiments. For inference, we used greedy decoding with temperature set to 0.0 for all models, including both baselines and simulations. We will release the code and data accompanying this work to facilitate replication and enable future research.

## ETHICS STATEMENT

In this work, we introduce ProbMedTOD, a task-oriented dialogue system for patient history taking. ProbMedTOD combines probabilistic reasoning using a BayesNet with conversational abilities of a large language model (LLM). The BayesNet parameters are estimated automatically using an LLM through self-consistency prompting. We evaluate our system on the publicly available MediTOD dataset, where it shows improved diagnostic reasoning and dialogue performance.

We reflect on the ethical aspects of our work along two dimensions: (1) potential deployment in real-world clinical settings, and (2) the resources used to build and evaluate ProbMedTOD.

**Deployment in Real-world Clinical Settings** We believe that ProbMedTOD represents a step towards building more reliable and interpretable dialogue systems for patient interaction. These qualities are important for earning the trust of medical professionals. However, we stress that our work is primarily a research project, motivated by the goal of combining traditional machine learning models with modern LLMs.

While it would be exciting to see systems like ProbMedTOD adapted for real-world use, such deployment requires careful consideration beyond diagnostic accuracy. For example, diseases may present differently based on a patient's demographics, lifestyle, or medical history. ProbMedTOD's probabilistic reasoning offers a foundation to handle such variations, but thorough evaluation by doctors across different demographics and specializations is essential to ensure patient safety.

ProbMedTOD is designed to systematically gather patient information. If used in practice, it is critical that any patient data collected complies with privacy and data protection laws, such as HIPAA. Additionally, patients must be clearly informed that they are interacting with an AI system, not a real doctor.

We also note that ProbMedTOD is not perfect and may make mistakes. These errors can result from LLM hallucinations, limitations in the BayesNet structure, inaccurate parameter estimations, or software bugs. Therefore, we strongly advise against using ProbMedTOD as a substitute for professional medical advice. Consulting a qualified doctor is essential for any health-related concerns.

**Resources Used in ProbMedTOD** We develop and evaluate ProbMedTOD using the MediTOD dataset, which is publicly available and contains annotated dialogues focused on pulmonary conditions. MediTOD is constructed through staged doctor-patient interactions by medical professionals, and no real patient data is involved.

As part of our work, we enhance MediTOD by annotating each dialogue with a diagnosis. This annotation process was carried out by paid professional annotators with a background in life sciences, under the supervision of a medical doctor. For building the BayesNet used in ProbMedTOD, we manually curate a disease-symptom structure based on publicly available medical resources. The BayesNet parameters are estimated using open-weight Gemma models. We plan to release these resources following the acceptance of our work to promote transparency and reproducibility.

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

## LIMITATIONS

While ProbMedTOD achieves strong results, it has several limitations. First, although we show improvements in supervised policy models, its effectiveness in in-context settings with LLMs remains unexplored. Second, our evaluation is restricted to the MediTOD dataset—the only publicly available English doctor–patient dialogue resource. Third, the Bayesian Network adopts a simplified bi-partite Noisy-OR design, capturing only direct symptom–disease links. In future, a more complete design involving additional nodes such as patient medical history, family history, habits and exposures can be explored. Further, demographics information could be incorporated to highlight ProbMedTOD's generalization. Finally, Bayesian Network structure is manually derived from documents, which poses scalability concerns. Future work could investigate automated construction using LLMs.

## A  DATASET DETAILS

Table 4 provides statistics for MediTOD dataset.

| | Train | Valid | Test |
|---|---|---|---|
| #Dialogues | 175 | 20 | 18 |
| #Utterances | 16852 | 1869 | 1798 |

Table 4: Statistics for MediTOD dataset.

## B  PROMPTS FOR BAYESNET ESTIMATIONS

---
**Disease Priors**

Based on your medical knowledge and experties regarding typical {{disease}} patterns, what is the estimated percentage likelihood that an otherwise healthy adult, with no known underlying health conditions will develop/contract acute bronchitis in a typical year? Use the following format to provide the answer. Outline your thinking briefly after you provide your answer.

<answer>write single number from 1 to 100, representing the percentage likelihood </answer>

---
**Symptom Likelihoods**

You are a medical professional and an expert in the field of medicine. You will be given a disease and a symptom. Your task is to predict the likelihood of the symptom given that a patient suffers from the disease. Base your prediction on your extensive knowledge of medical literature and clinical experience. Specifically, provide a number between 0-100 indicating the likelihood. Enclose your answer in `<answer>` and `<answer>` tags. For example: <answer>55 <answer>

Here is the disease and the symptom for which you need to predict the likelihood.

# Disease
{{disease}} (in its typical presentation in adult patients)

# Symptom
{{symptom}} (as reported by the patient)

Outline your thinking briefly before you provide your answer.

---

## C  EXACT BAYESIAN INFERENCE USING NOISY-OR

Following (Guan & Baral, 2021; CMS, 2024), we assume a primary diagnosis model, i.e., chiefly one disease is responsible for the patient's condition. To compute the posterior $P(D_j = 1 \mid \mathcal{S}^+, \mathcal{S}^-)$, we start with marginalizing over all $D_{j-}$ combinations.

$$P(D_j = 1 \mid \mathcal{S}^+, \mathcal{S}^-) = \sum_{\forall D_{j-}} P(D_j = 1, D_{j-}|\mathcal{S}^+, \mathcal{S}^-)$$
$$= P(D_j = 1, D_{j-} = 0|\mathcal{S}^+, \mathcal{S}^-)$$

Now, posterior is computed by applying Bayes' Rule.

$$P(D_j = 1 \mid \mathcal{S}^+, \mathcal{S}^-)$$
$$= \frac{P(\mathcal{S}^+\mathcal{S}^-|D_j = 1, D_{j-} = 0)P(D_j = 1, D_{j-} = 0)}{P(\mathcal{S}^+, \mathcal{S}^-)}$$

Note that the term $P(\mathcal{S}^+, \mathcal{S}^-)$ is common for all diseases $D_j$. Applying conditional independence of symptoms given disease, we compute the posterior using the following equation.

> **Gold Diagnosis - Bronchitis  Predicted Diagnosis - Bronchitis**
>
> Patient: oh, hello doctor. i've come in because i've had a persistent cough and i've been feeling very weak.
> Doctor: ok, and when did the cough start?
> Patient: it started a while ago, doctor. i can't quite recall the exact date, but it's been going on for some time now.
> Doctor: ok, and how long is a while?
> Patient: oh, dear. it's hard to say exactly, doctor. it feels like it's been a few weeks, maybe longer. it's been going on for quite a while now.
> Doctor: ok, and is it a dry cough or a cough with mucus?
> Patient: oh, it's not dry, doctor. i've been coughing up some phlegm.
> Doctor: ok, and what color is the phlegm?
> Patient: oh, it's sort of grayish-green, doctor.
> Doctor: ok, and have you noticed any blood?
> Patient: no, doctor, i don't think so. i haven't seen any blood.
> Doctor: ok, and has the cough been getting worse or staying the same?
> Patient: well, doctor, it hasn't really gotten better. i've been coughing quite a bit, and i'm still feeling weak.
> Doctor: ok, and is there anything that makes the cough worse?
> Patient: well, doctor, sometimes when i lie down, i feel like i can't catch my breath, and the coughing gets worse then. it's just hard to breathe sometimes.
> Doctor: ok, and when did the shortness of breath start?
> Patient: oh, the shortness of breath? i think it started around the same time as the cough, doctor. it's been getting worse lately, especially when i try to lie down.
> Doctor: ok, and do you find that you wake up in the middle of the night to take a breath?
> Patient: oh, waking up at night? no, i don't think so, doctor. i just feel tired, especially when i lie down.
> Doctor: ok, and how is your energy been?
> Patient: oh, doctor, my energy is very low. i've been feeling quite weak, especially lately with this cough. ... ...

$$P(D_j = 1 \mid \mathcal{S}^+, \mathcal{S}^-) \propto \ P(D_j = 1, D_{j^-} = 0)$$
$$\times \prod_{i \in \mathcal{S}^+} P(S_i = 1 \mid D_j = 1, D_{j^-} = 0)$$
$$\times \prod_{i \in \mathcal{S}^-} P(S_i = 0 \mid D_j = 1, D_{j^-} = 0)$$

where $P(D_j = 1, D_{j^-} = 0) = P(D_j = 1) \prod_{\forall j' \neq j} (1 - P(D_{j'} = 1))$ are the priors.

## D  EXAMPLE DIALOGUE GENERATED BY PROBMEDTOD (QWEN3-8B)

## E  SIMULATION DETAILS

### E.1  NLU AND NLG MODELS

Both NLU and NLG models are LLaMA3 8B models fine-tuned on the respective task data from MediTOD. At each dialogue turn, NLU model inputs the entire dialogue history and the last doctor action to predict intent and slot-value pairs from the latest patient utterance. NLG model inputs the entire dialogue history and the predicted doctor action to generate doctor response. Our NLU model operates at 86.57% test Medical F1 and NLG model operates at 20.59 test BLEU score showcasing high performance.

**Patient Response to Inquire Actions**

In the conversation given below, the doctor is interviewing a participant (either patient or a guardian in case the patient is a child). In his last utterance, the doctor is making an inquiry. Assume the role of the participant and respond to the doctor's inquiry. Specifically, convert the given answer sketch into an appropriate patient response to the doctor. Instructions for formulating the patient's response are given below.

# Task Instructions (procedure to formulate the answer)
1. Read the last doctor utterance. Analyze what information is being inquired in the last utterance.
2. Determine whether the participant is a patient or a guardian. If the participant is a guardian, assume the role of the guardian and respond to the doctor's inquiry.
3. Convert the answer sketch into an appropriate response to the doctor. Make sure that the response is meaningful and relevant to the inquiry.
4. Enclose the response in `<answer>` and `</answer>` tags.
5. The response must be TRUTHFUL to the answer sketch. The response MUST NOT INCLUDE any information not present in the answer sketch. The response must be honest, meaningful, and concise.
6. Remember you are not a medical expert. Your response must use layman's language devoid of any medical terms.
7. The response must continue the conversation fluently and meaningfully.
8. The response must be SHORT preferably a few words. It should not include any unnecessary information.
9. If the answer sketch is empty, your response must indicate that you cannot provide any information.
Describe your thinking for EACH of the above 8 steps in details.

# Output Format
**thinking**: <your detailed thinking for each of the 8 steps>
**output**: <your final answer enclosed in <answer>and </answer>tags>

# Doctor-patient Conversation
{{dialogue_history}}

# Last Doctor Utterance (a question you need to respond to.)
{{last_doctor_uttr}}

# Answer sketch (a sketch for your answer - includes information to be used for answering the doctor's query)
```JSON
{{answer_sketch}}
```

E.2 PATIENT RESPONSE GENERATION PROMPTS

**Patient Response to Chit-chat Actions**

In the conversation given below, the doctor is interviewing a participant (either patient or a guardian in case the patient is a child). In his last utterance, the doctor is making a casual conversation. Assume the role of the participant and respond to the doctor's chit-chat. Keep the response short and casual. Enclose the response in `<answer >` and `</answer >` tags. Here is the doctor-patient conversation.

# Doctor-patient Conversation
{{dialogue_history}}

# Last Doctor Utterance (a question you need to respond to.)
{{last_doctor_uttr}}

**Patient Response to Diagnosis**

In the conversation given below, the doctor is interviewing a participant (either patient or a guardian in case the patient is a child). In his last utterance, the doctor is making a diagnosis. Assume the role of the participant and respond to the doctor's diagnosis. Accept the doctor's diagnosis as FINAL, regardless of the ongoing dialogue. Do not contradict the doctor. Continue the conversation by simply ACKNOWLEDGING and ACCEPTING the diagnosis. Keep the response short. Enclose the response in `<answer>` and `</answer>` tags. Here is the doctor-patient conversation.

# Doctor-patient Conversation
{{dialogue_history}}

# Last Doctor Utterance (a question you need to respond to.)
{{last_doctor_uttr}}

**Patient Response to Salutations**

In the conversation given below, the doctor is interviewing a participant (either patient or a guardian in case the patient is a child). In his last utterance, the doctor is concluding the conversation. Assume the role of the participant and respond to the doctor's conclusion. Express gratitude for the consultation. Keep the response short. Enclose the response in `<answer>` and `</answer>` tags. Here is the doctor-patient conversation.

# Doctor-patient Conversation
{{dialogue_history}}
# Last Doctor Utterance (a question you need to respond to.)
{{last_doctor_uttr}}

