# OpenReview forum: "ProbMedTOD: A Bayesian Network Guided Task-Oriented Dialogue System for Patient History Taking"
_ICLR.cc/2026/Conference — Submitted to ICLR 2026_

### Official Review · Reviewer_aWD2 · 2025-11-01

**Soundness:** 2
**Presentation:** 2
**Contribution:** 2
**Rating:** 4
**Confidence:** 4

**Summary:**

The paper proposes ProbMedTOD, a task-oriented medical dialogue system that combines an explicit disease-symptom Bayesian Network with an LLM-based policy. At each turn, an NLU module extracts structured symptoms, a Noisy-OR BayesNet updates posterior probabilities over candidate principal diagnoses, and the policy LLM uses those posteriors together with the dialogue state to choose the next question. BayesNet parameters, namely disease priors and symptom likelihoods, are estimated directly by an LLM via self-consistency prompting, which avoids reliance on patient-level data. For evaluation, the authors build a patient simulator that constructs profiles from MediTOD and MIMIC-IV, then uses an LLM to generate faithful patient responses for multi-turn interactions. Relative to supervised LLM baselines and a retrieval-augmented baseline, ProbMedTOD shows modest gains on turn-level Medical-F1 and Precision@K and larger gains on dialogue-level diagnostic ranking measured by MRR and Hit@K. An ablation that removes the BayesNet degrades performance, indicating that the explicit probabilistic layer contributes to the gains.

**Strengths:**

(a) The method injects explicit uncertainty modeling by feeding Bayesian posteriors to the policy, which encourages questions that reflect diagnostic likelihoods rather than treating hypotheses uniformly.

(b) The parameter estimation procedure is practical in data-constrained settings, since self-consistency with an LLM provides priors and likelihoods without patient-level annotation.

(c) The evaluation focuses on the diagnostic objective, using dialogue-level ranking metrics within a simulator that rewards information gathering rather than surface action matching.

(d) The BayesNet scaffold provides an interpretable structure whose posteriors can in principle be audited by clinicians.

**Weaknesses:**

(a) The novelty is limited relative to prior Bayesian inquiry frameworks. The main addition is the combination of a classical probabilistic scaffold with a modern LLM policy and LLM-based parameter estimation.

(b) Validation of LLM-estimated probabilities is thin. The paper does not compare priors and likelihoods with epidemiology or expert labels, and the reported sanity check is indirect.

(c) The single-disease assumption simplifies inference but departs from clinical reality with comorbidities, which can bias posteriors and question selection.

(d) The realism of the simulator is uncertain. LLM-generated patients may be overly cooperative or align too closely with profiles, which can inflate reported gains.

(e) Baselines are not sufficiently strong. Missing comparisons include information-gain or RL-based policies, RAG with CoT, and state-of-the-art closed models.

(f) There is no analysis of calibration, stopping rules, or escalation policies, all of which are crucial in medical applications.

(g) The engineering and inference overhead is not weighed against the relatively modest gains on turn-level metrics, and no cost-efficiency analysis is provided.

**Questions:**

(a) Can you validate the LLM-estimated priors and likelihoods against expert annotations or published statistics, even on a small subset, to bound estimation error.

(b) How would you extend inference beyond a single principal diagnosis, for example to a sparse multi-label posterior with approximate inference and k greater than one active diseases.

(c) Can you add stronger baselines such as information-gain policies, RAG with CoT (IRCoT, Self-RAG, ...), or a modern closed-model policy in order to clarify the performance gap.

(d) Do you have any human evaluation by clinicians of question quality and ordering or case-level diagnostic utility.

(e) How sensitive are the results to the parameter-estimation procedure, for example fewer self-consistency samples, different estimation LLMs, or uniform priors.

(f) Are posteriors calibrated, and do you have principled stopping or deferral rules when uncertainty remains high.

(g) How well does the approach transfer beyond pulmonary cases, for example on a held-out specialty or an out-of-distribution symptom mix.

(h) How robust are the results if patient responses are noisy or contradictory or generated by a different LLM, and do adversarial perturbations change your conclusions.

---

> ### Author Response · Authors · 2025-12-03
>
> We appreciate the time and effort you have taken to review our work. Please find our response to your concerns below.
>
> **The novelty is limited relative to prior Bayesian inquiry frameworks. The main addition is the combination of a classical probabilistic scaffold with a modern LLM policy and LLM-based parameter estimation.**
>
> We respectfully note that our work addresses two important limitations in current medical dialogue systems. First, existing systems often rely on pattern matching and do not explicitly handle the uncertainty present in diagnostic reasoning. ProbMedTOD addresses this by incorporating a Bayesian Network into the dialogue policy, supporting decision-making under uncertainty. Second, we reduce the need for large datasets or manual engineering in probabilistic modeling by introducing a method to estimate parameters directly from LLMs using structured self-consistency prompting. We validate this approach through simulations with MediTOD and MIMIC-IV profiles, where it outperforms standard LLM and RAG baselines on diagnostic tasks.
>
> **Validation of LLM-estimated probabilities is thin. The paper does not compare priors and likelihoods with epidemiology or expert labels, and the reported sanity check is indirect.**
>
> We acknowledge the importance of validating estimated probabilities against established epidemiological data. Direct validation on the MediTOD dataset is challenging as it lacks ground-truth annotations for disease priors or symptom likelihoods.
>
> However, to address this concern, we have performed an external evaluation using the NLICE dataset, which provides ground-truth probability tables for direct comparison. We found a strong correlation between our LLM-estimated parameters and the NLICE ground truth. Please refer to our Response to Reviewer TTFU for details.
>
> **The single-disease assumption simplifies inference but departs from clinical reality with comorbidities, which can bias posteriors and question selection.**
>
> We agree that assuming a single disease is a simplification, as real patients often have comorbidities. This assumption was made mainly to keep exact posterior inference tractable and parameter estimations feasible.
>
> However, the ProbMedTOD is not inherently limited to this assumption. The BayesNet structure can be extended to support multiple concurrent diagnoses without changing the core design. Our response to Reviewer TTFU discusses an possible extension that can be implemented Response to Reviewer TTFU.
>
> **The realism of the simulator is uncertain. LLM-generated patients may be overly cooperative or align too closely with profiles, which can inflate reported gains.**
>
> We appreciate this observation regarding simulator fidelity. LLM-based patient simulators tend to be cooperative, following the provided profiles closely and answering questions fully. Introducing patient personalities that make errors in reporting or withhold information would be an interesting way to test system robustness.
>
> While this was not the primary focus of our work, both ProbMedTOD and the baseline models (RAG, vanilla LLMs) were evaluated under the same simulation conditions, ensuring a fair comparison of diagnostic strategies. Exploring unreliable or partially informative patients could be a valuable direction for future work.
>
> **Baselines are not sufficiently strong. Missing comparisons include information-gain or RL-based policies, RAG with CoT, and state-of-the-art closed models.**
>
> To address your concern, we have conducted a new set of experiments using Gemini 2.5 Flash. We evaluated this model in three configurations: a) state-of-the-art foundation models (Gemini 2.5 Flash, GPT-5 Mini) with CoT prompting, b) Gemini 2.5 Flash \+ RAG: A system combining an SOTA CoT model with retrieval, and b) ProbMedTOD (Gemini 2.5 Flash) \- Integrating our probabilistic framework with the Gemini. For the detailed results table and analysis, please refer to our response to Reviewer mTGg.
>
> **There is no analysis of calibration, stopping rules, or escalation policies, all of which are crucial in medical applications.**
>
> The main goal of our work is to improve history taking and diagnostic reasoning in medical dialogue systems. For stopping criteria, our model follows a learned policy: it ends the session either when it predicts the Salutation action (similar to dialogues in MediTOD) or when it reaches a maximum number of turns. We agree that studying calibration and escalation policies is important for real-world use. We see this as future work, noting that ProbMedTOD’s probabilistic outputs ($P(D|S)$) provide a starting point to add these safety measures.

---

> > ### Author Response · Authors · 2025-12-03
> >
> > **The engineering and inference overhead is not weighed against the relatively modest gains on turn-level metrics, and no cost-efficiency analysis is provided.**
> >
> > We measure the inference overhead on the MediTOD test set of 877 samples using a single A100 GPU. The results are summarized below:
> >
> > | Model | Number of Tokens Processed | Processing Time per Sample |
> > | :---- | :---- | :---- |
> > | Llama3 8B | 1.1M | 0.106 |
> > | ProbMedTOD (Llama3 8B) | 6.5M | 0.159 |
> >
> > The inference time increases by only 53ms per turn. This ensures ProbMedTOD remains fully viable for real-time clinical interactions.
> >
> > We acknowledge a ≈6x increase in token consumption. This is expected, as ProbMedTOD augments the prompt with structured diagnostic hypotheses (BayesNet outputs) at every turn. While the token cost is higher, we argue this is a favorable trade-off for the improvement in MRR.
> >
> > **How would you extend inference beyond a single principal diagnosis, for example to a sparse multi-label posterior with approximate inference and k greater than one active diseases.**
> >
> > The Noisy-OR model can naturally compute the posterior for any combination of diseases, without relying on a single-disease assumption (Jordan et al.). However, calculating the exact posterior in this case can be intractable. Approximate inference methods, such as variational inference, could be used to address this. Alternatively, adding a co-morbidity structure to the BayesNet can simplify computation. In our response to Reviewer TTFU, we show one such variation.
> >
> > Predicting only one diagnosis is not a limitation for ProbMedTOD. Instead of using the primary diagnosis from beam search, the system can be adapted to predict multiple diagnoses. Specifically, we modify ProbMedTOD to output its final diagnostic belief states rather than a single label at the end of the dialogue, which significantly improves its diagnostic performance.
> >
> > We appreciate your insightful suggestions and have revised our results accordingly. Please refer to our general comment for the updated results.
> >
> > Reference:
> >
> > * *Jaakkola, Tommi S., and Michael I. Jordan. "Variational probabilistic inference and the QMR-DT network." Journal of artificial intelligence research 10 (1999): 291-322.*
> >
> > **How sensitive are the results to the parameter-estimation procedure, for example fewer self-consistency samples, different estimation LLMs, or uniform priors.**
> >
> > Regarding the sensitivity of results to the parameter-estimation procedure, such as using fewer self-consistency samples, different estimation LLMs, or uniform priors, please see our response to Reviewer TTFU. There, we evaluate the quality of predicted probabilities against the gold-standard NLICE BayesNet. We find that estimates from LLMs such as Gemma2 27B and Qwen3 32B achieve lower average KL Divergence than uniform probabilities, indicating that the predictions are meaningful. Additionally, increasing the number of self-consistency samples from 10 to 50 further improves probability quality.
> >
> > **How well does the approach transfer beyond pulmonary cases, for example on a held-out specialty or an out-of-distribution symptom mix.**
> >
> > Please refer to our response to Reviewer TTFU.
> >
> > **Do you have any human evaluation by clinicians of question quality and ordering or case-level diagnostic utility.**
> >
> > Please refer to our response to Reviewer mTGg.

---

### Official Review · Reviewer_noDp · 2025-11-01

**Soundness:** 3
**Presentation:** 3
**Contribution:** 2
**Rating:** 6
**Confidence:** 3

**Summary:**

The paper proposes ProbMedTOD, a task-oriented dialogue system that integrates a disease–symptom Bayesian Network with an LLM-based dialogue policy for patient history-taking. The BayesNet updates diagnostic probabilities at each turn, which guide the question generation by the policy model. The approach is evaluated on MediTOD and MIMIC-IV datasets and shows improvements over RAG and LLM-only baselines in both turn-level and dialogue-level metrics. The method is conceptually interesting and well-motivated, though some aspects of the evaluation and interpretability remain underdeveloped.

**Strengths:**

# originality
The integration of explicit probabilistic reasoning into a dialogue policy framework is novel and well-motivated, bridging the gap between intuitive (System 1) and deliberative (System 2) reasoning in medical settings. The approach can be generalized to other domains involving diagnostic questioning or hypothesis-driven dialogue.

# quality

The methodology is sound and clearly implemented. The ablations demonstrate that the Bayesian component makes a meaningful contribution to the results. Although the BayesNet is not large, its incorporation provides a principled approach to uncertainty reasoning and enhances diagnostic focus.

# clarity
The paper is clear in its chosen architecture.

# significance
The problem considered is relevant and such a system can have an important impact on healthcare.

**Weaknesses:**

# originality
The paper could explore interpretability. This is mentioned in the paper but never explored. Additionally, other methods of encoding the output of the BayesNet should be tested, as LLMs are known for not being the best at handling numerical inputs. For example, an ordered list of the BN probabilities or semantic tags for a probability range should be tested.

# quality
Aspects missing from the evaluation:
1. Evaluation with clinical experts would be beneficial.
1. An ablation assessing the impact of errors in the BayesNet.
1. Tables would benefit from the indication of stdevs
1. Impact of fine-tuning of the policy model on MediTOD. This information could be useful for other implementations.


The value of Med F1 for ProbMedTOD with the Qwen model in Table 3 is different from the homologous value in Table 1 (and largely so). Can the authors explain this?

# clarity
The paper is clear. Metrics should not be presented only as acronyms.


Future work could explore reinforcement learning or information–gain–based policies built on the same probabilistic backbone, and extend the system to multi-disease reasoning or multimodal evidence (e.g., lab tests). Expert-based validation of dialogues would be a key next step toward clinical applicability.

**Questions:**

See weaknesses

---

> ### Author Response · Authors · 2025-12-03
>
> We appreciate your time and insightful feedback. Please find below our reply to your concerns.
>
> **Evaluation with clinical experts would be beneficial.**
>
> Please refer to our response to Reviewer mTGg.
>
> **The paper could explore interpretability.**
> We appreciate the suggestion to expand on interpretability. In the final version, we will add a dedicated section on *interpretable diagnostic reasoning* in ProbMedTOD. One advantage of our probabilistic design is that it naturally showcases the model’s evolving diagnostic beliefs at each turn, allowing readers to trace how specific patient responses influence subsequent questions and belief updates.
>
> To illustrate this, we provide a sample dialogue from a MIMIC-based simulation using ProbMedTOD (Llama3-8B) where the true diagnosis is Pulmonary Fibrosis. The example shows how the system (1) starts with chief complaints and asks clinically meaningful questions, (2) updates its disease posterior whenever a new symptom is uncovered, and (3) narrows down its differential diagnosis in a transparent and interpretable manner. As new symptoms are revealed, such as dry cough without sputum, the model correspondingly elevates Pulmonary Fibrosis and deprioritizes diseases like TB or AE-COPD, demonstrating a clear link between evidence and belief revision. Disease posteriors are specified in brackets.
>
> | Turn | Speaker | Utterance | Diagnostic Hypothesis | Comment |
> | ----- | :---- | :---- | :---- | :---- |
> | 1 | Patient | I've been feeling really short of breath lately, especially when I walk up the stairs. | AE-COPD (31); RDS (26); Pulmonary fibrosis (13); TB (8); Pneumothorax (6) | Initial broad differential based on dyspnea. |
> | 2 | Doctor | OK, and when did this start? |  | The doctor gathers additional details about the patient's chief complaint. |
> | 3 | Patient | It started about two weeks ago, I think. It's been getting worse since then. |  |  |
> | 4 | Doctor | OK, is it just the walking up the stairs that's short of breath, or are you getting short of breath with other activities as well? |  |  |
> | 5 | Patient | Oh, it's not just the stairs, doctor. Even just walking around the house, or doing simple things, I get winded pretty easily now. |  |  |
> | 6 | Doctor | OK, and when did that start? |  |  |
> | 7 | Patient | Oh, that started around the same time, maybe a little after the main shortness of breath began. So, about two weeks ago, I guess. |  |  |
> | 8 | Doctor | OK, and is there anything that makes it better or worse? |  |  |
> | 9 | Patient | Not really, doctor. It just seems to be getting worse overall, especially when I try to do anything. |  |  |
> | 10 | Doctor | OK, do you have a cough? |  | Inquiring about coughs refines respiratory differential. |
> | 11 | Patient | Yes, I have a cough. It's been going on for about two weeks too, and it's a dry cough. I haven't been coughing anything up. | AE-COPD (51); Pulmonary fibrosis (20); TB (12); Bronchitis (5); Emphysema (3) | Model incorporates cough and adjusts probabilities. |
> | 12 | Doctor | OK, so no sputum? |  | Sputum type helps differentiate TB vs fibrosis. |
> | 13 | Patient | Yes, doctor. It's just a dry cough, nothing coming up. | Pulmonary fibrosis (63); Emphysema (10); Bronchospasm (7); CHF (6); COVID-19 (5) | Dry cough elevates fibrosis; TB drops from top 5\. |
> | 14 | Doctor | OK, any blood? |  |  |
> | 15 | Patient | No, doctor. No blood. | Pulmonary fibrosis (64); Emphysema (10); Bronchospasm (7); CHF (6); COVID-19 (5) | Lack of hemoptysis lowers suspicion of TB/infection. |
> | 16 | Doctor | OK, any wheezing? |  |  |
> | 17 | Patient | No, doctor, I haven't noticed any wheezing. |  |  |
>
> This example highlights how ProbMedTOD’s structure enables *step-wise interpretability*: each question can be linked to a measurable belief update, and each belief update corresponds to specific evidence in the dialogue. We will integrate such interpretability demonstrations and discussion into the final manuscript.
>
> **Additionally, other methods of encoding the output of the BayesNet should be tested, as LLMs are known for not being the best at handling numerical inputs. For example, an ordered list of the BN probabilities or semantic tags for a probability range should be tested.**
>
> We tested whether using a semantic probability scale would improve alignment compared to our numerical approach. Using the NLICE dataset, we prompted the model to give five semantic labels—*Highly Unlikely, Unlikely, Moderately Likely, Likely,* and *Highly Likely*—instead of numeric probabilities. We then mapped these labels to discrete probability values and measured the average KL Divergence against the ground truth.
>
> The semantic scale produced a KL Divergence of 0.463, while our numerical method achieved 0.234. This shows that the coarse semantic representation reduced probabilistic accuracy, whereas direct numerical estimates aligned much more closely with the true medical distributions.

---

> > ### Author Response · Authors · 2025-12-03
> >
> > **Tables would benefit from the indication of stdevs**
> >
> > Although our current results use greedy decoding (temperature \= 0.0) to ensure deterministic reproducibility, we agree that reporting variance is important. We will re-run experiments with multiple training seeds and will include standard deviations in the final version of the paper.
> >
> > **Impact of fine-tuning of the policy model on MediTOD. This information could be useful for other implementations.**
> >
> > To quantify the impact of fine-tuning on MediTOD, we evaluate our fine-tuned policy model (ProbMedTOD Llama3-8B) against the base Llama3-8B model on standard general reasoning benchmarks (GSM8K (Karl, et.al.) and ARC-Challenge (Etzioni, O., et. al.)). The results are presented below:
> >
> > | Model | GSM8K | ARC-Challenge |
> > | :---- | :---- | :---- |
> > | Llama3 8B | 0.757 | 0.531 |
> > | ProbMedTOD (Llama3 8B) | 0.689 | 0.493 |
> >
> > Fine-tuning on the specialized MediTOD dataset results in a regression in general reasoning capabilities. This trade-off, often attributed to catastrophic forgetting, is expected when optimizing smaller models for highly specific medical protocols without replay buffers. To maintain general reasoning of the model, a simple strategy of mixing instruction tuning dataset with MediTOD can be explored in the future.
> >
> > References:
> >
> > * *Cobbe, Karl, et al. "Training verifiers to solve math word problems." arXiv preprint arXiv:2110.14168 (2021).*
> > * *Clark, P., Tafjord, O., and Etzioni, O. "Think you have Solved Question Answering? Try ARC, the AI2 Reasoning Challenge". arXiv preprint arXiv:1803.05457, 2018\.*
> >
> > **The value of Med F1 for ProbMedTOD with the Qwen model in Table 3 is different from the homologous value in Table 1 (and largely so). Can the authors explain this?**
> >
> > We apologize for the clerical error in Table 3\. We confirm that the value reported in **Table 1 (0.273) is the correct Medical F1 score** for ProbMedTOD (Qwen3 8B). We have corrected the table below to reflect the accurate ablation results.
> >
> > | Model | MediTOD |  | MIMIC-IV (297 cases) |  |
> > | :---- | :---- | :---- | :---- | :---- |
> > |  | Med F1 | Precision@Inf | MRR | Hit@5 |
> > | ProbMedTOD (Qwen3 8B) | 0.273 | 0.453 | 0.311 | 0.529 |
> > | w\\o BayesNet | 0.268 | 0.436 | 0.168 | 0.374 |
> >
> > The corrected results consistently show that while the turn-level *Medical F1* difference is modest (0.273 vs 0.268), the *diagnostic metrics* (MRR and Hit@5) suffer a significant drop without the BayesNet. We will update the final paper with these correct values.

---

### Official Review · Reviewer_mTGg · 2025-11-02

**Soundness:** 2
**Presentation:** 3
**Contribution:** 3
**Rating:** 6
**Confidence:** 3

**Summary:**

This paper proposes PROBMEDTOD, a Bayesian Network–guided Task-Oriented Dialogue (TOD) system designed to assist with structured patient history taking. The system integrates a Bayesian Network (BN) for modeling probabilistic dependencies between clinical symptoms and conditions, guiding the dialogue agent’s question-asking strategy. Unlike conventional medical dialogue systems that rely solely on goal-oriented policies or LLMs, PROBMEDTOD aims to reduce unnecessary or redundant questioning, improve clinical relevance, and support more accurate patient assessment. Experiments on a simulated clinical dialogue environment show improvements in diagnostic questioning efficiency and task success rates over baselines.

**Strengths:**

- The idea of incorporating structured probabilistic reasoning into a medical TOD agent is theoretically sound and interesting for medical AI.
- he hybrid design leverages BN inference to guide the dialogue policy, which is well-justified for medical settings where causal and comorbidity relationships matter. This is conceptually sound and aligns with clinical exam logic, which often reasons over conditional probabilities rather than text similarity alone.
- The paper is generally easy to follow, with intuitive diagrams (e.g., the BN-TOD architecture figure) that help communicate the workflow.

**Weaknesses:**

- Evaluation is conducted only in a simulated setting. Although this is understandable due to data constraints, real-patient or clinician-in-the-loop evaluation would significantly strengthen the claims of clinical relevance.
- Baseline comparisons are limited. The evaluation omits comparisons with more recent LLM-based medical agents or retrieval-augmented systems. Given rapid advances in medical LLMs, comparison only with classic TOD baselines understates the challenge. The paper also does not justify the absence of stronger baselines.

**Questions:**

- How was the BN constructed and validated? Was any external clinical source (e.g., medical knowledge graph, clinician review) used to ensure correctness?
- Why were modern LLM-based medical dialogue systems not included as baselines? Even small-scale comparisons would strengthen the contribution.
- Have you considered evaluating with medical professionals (even small-scale) to validate usefulness, safety, or alignment with history-taking best practices?

---

> ### Author Response · Authors · 2025-12-03
>
> We appreciate your time and insightful review. Please find our response to your queries below.
>
> **Evaluation is conducted only in a simulated setting. Although this is understandable due to data constraints, real-patient or clinician-in-the-loop evaluation would significantly strengthen the claims of clinical relevance.**
>
> We agree that clinician-in-the-loop evaluation is the gold standard. Although we were unable to complete our planned human-evaluation study, we performed an LLM-as-a-Judge assessment using Gemini-2.5-Pro and evaluated over 100 randomly sampled patient cases from the MIMIC test set. Our Clinical Dialogue Evaluation Protocol scores models along three axes:
>
> * History Taking: Logical, complete, and non-repetitive information gathering.
> * Evidence Sufficiency: Whether the “Gold Truth’’ diagnosis appears among the top candidates.
> * Clinical Utility: Practical usefulness in a real clinical workflow.
>
> Results (1–5 scale):
>
> | Model | History | Evidence | Utility |
> | ----- | ----- | ----- | ----- |
> | ProbMedTOD (Ours) | 3.02 | 3.48 | 2.96 |
> | Llama3-8B (FT) | 2.96 | 3.46 | 2.93 |
> | RAG (Llama3-8B) | 2.89 | 3.36 | 2.85 |
>
> ProbMedTOD leads across all metrics. Its higher History Taking score reflects the benefit of the Bayesian structure for guiding targeted questioning. This translates into stronger Evidence Sufficiency and improved Clinical Utility, indicating fewer gaps that would force clinicians to re-interview.
>
> While full clinician-in-the-loop validation remains future work, our Gemini-based evaluation on 100 real MIMIC cases shows that ProbMedTOD delivers clear gains in dialogue quality, diagnostic reasoning, and practical utility.
>
> **Baseline comparisons are limited. The evaluation omits comparisons with more recent LLM-based medical agents or retrieval-augmented systems. Given rapid advances in medical LLMs, comparison only with classic TOD baselines understates the challenge. The paper also does not justify the absence of stronger baselines.**
>
> We appreciate your feedback and have conducted additional experiments to address it. Our initial choice of baselines was driven by the specific constraints of the supervised Task-Oriented Dialogue (TOD) setting, where models are typically fine-tuned on domain-specific dialogue/state pairs.
>
> To provide a comparison against SOTA foundation models, we ran simulations using Gemini 2.5 Flash and GPT-4.1-Mini on the expanded MIMIC test set, where both models decide the next doctor question with CoT prompting. We also experiment with a hybrid approach, combining RAG (Gemini 2.5 Flash) and ProbMedTOD (Gemini 2.5 Flash), where policy decisions are guided by retrieval and BayesNet outputs, respectively. Additionally, we compare our approach with the Information Gain policy, which selects the next symptom to ask in order to maximize the reduction in disease posterior entropy. However, the final response generation utilizes Gemini's reasoning. Results are as follows:
>
> | Model | MRR | Hit@1 | Hit@3 | Hit@5 |
> | :---: | :---: | :---: | :---: | :---: |
> | RAG (Llama3 8B) | 0.175 | 0.077 | 0.216 | 0.384 |
> | ProbMedTOD (Llama3 8B) | 0.263 | 0.152 | 0.354 | 0.485 |
> | ProbMedTOD (Qwen 8B) | 0.311 | 0.202 | 0.387 | 0.529 |
> | Information Gain | 0.268 | 0.185 | 0.323 | 0.421 |
> | GPT-4.1-Mini | 0.513 | 0.407 | 0.613 | 0.670 |
> | Gemini 2.5 Flash | 0.513 | 0.411 | 0.609 | 0.667 |
> | Gemini 2.5 Flash (RAG) | 0.436 | 0.340 | 0.546 | 0.572 |
> | Gemini 2.5 Flash (ProbMedTOD) | 0.475 | 0.381 | 0.566 | 0.613 |
>
> Our results show that Gemini 2.5 Flash and GPT-4.1-Mini perform strongly (MRR \~0.51). However, being proprietary models, they may have some unknown overlap with public medical datasets. Their use in clinical settings is also limited by patient privacy requirements and dependence on external APIs.
>
> In practice, smaller models that can be hosted locally are preferable to ensure privacy and control. In this setting, ProbMedTOD (Qwen 8B) substantially outperforms standard baselines and the Information Gain model, achieving nearly twice the MRR of RAG with Llama3 8B (0.311 vs. 0.175). Even with larger foundation models, ProbMedTOD (Gemini) outperforms the RAG (Gemini) baseline (0.475 vs. 0.436 MRR).
>
> We note a gap between the unconstrained Gemini model (0.513 MRR) and ProbMedTOD (0.475 MRR), likely due to ProbMedTOD’s fixed ontology. This points to a future direction: leveraging LLMs to dynamically expand the probabilistic structure, combining the safety of fixed schemas with greater flexibility.

---

> > ### Author Response · Authors · 2025-12-03
> >
> > **How was the BN constructed and validated? Was any external clinical source (e.g., medical knowledge graph, clinician review) used to ensure correctness?**
> >
> > We built the BayesNet structure using MedlinePlus (https://medlineplus.gov/). For each of the 34 diseases in the MediTOD dataset, we automatically extracted the associated symptoms from their respective MedlinePlus pages. This ensures that all disease–symptom links come from verified medical sources rather than model-generated assumptions.
> >
> > For validation, we manually corrected syntactic issues during the construction process. We also evaluated the quality of our learned parameters by comparing them with the established NLICE medical Bayesian Network. Please refer to our response to reviewer TTFU.

---

### Official Review · Reviewer_TTFU · 2025-11-04

**Soundness:** 3
**Presentation:** 2
**Contribution:** 3
**Rating:** 4
**Confidence:** 4

**Summary:**

The manuscript proposes ProbMedTOD, a hybrid medical task-oriented dialogue system integrating a disease-symptom Bayesian Network  for probabilistic diagnostic reasoning, and a supervised policy LLM conditioned on posterior disease probabilities. To evaluate diagnostic success, the authors introduce a patient simulator with cases from MediTOD and MIMIC-IV. Experiments show improvements over LLM baselines and RAG approaches in both turn-level metrics and diagnostic performance.

**Strengths:**

- The paper introduces a practical and efficient method to estimate BayesNet parameters using structured LLM prompts with self-consistency.
- Strong conceptual grounding: the paper clearly frames the problem as bridging the gap between System 1 intuitive LLM fluency and System 2 deliberate clinical reasoning.
- The system outperforms LLM and RAG baselines on both turn-level metrics and dialogue-level simulation metrics. Extra ablation study show the effectiveness of the BayesNet module.

**Weaknesses:**

- Limited clinical coverage and scalability concerns: the evaluation contains only to pulmonary MediTOD and small MIMIC-IV subset. This limitsgeneralizability to other medical fields which may have different reasoning patterns and disease/symptom structures.
- In addition, the BayesNet structure was constructed via programmatic extraction followed by manual validation, where not reproducibly specified and the manual step also limits the system's applicability to new medical domains.
- Validation relies on comparing primary and other edges' likelihood averages, rather than benchmarking against an external ground truth using real-world epidemiological data or clinical expert consensus. The risk of the LLM encoding inaccurate or biased associations remains unanalyzed.
- Eq. (2) references “Equation 2.1” but equation numbering is inconsistent (Sec. 3).

**Questions:**

- The core inference simplification (Eq. 2)  is central of the method to make Bayesian inference tractable. However, patients in real-world clinical settings often have multiple concurrent diagnoses (comorbidities), which this model cannot naturally handle. Although model claims extensibility, is there any evidence to support this?
- The model sometimes asks non-informative questions (Sec. 7), is there any systematic analysis or mitigation strategy?

---

> ### Author Response · Authors · 2025-12-02
>
> We sincerely appreciate your careful review and valuable feedback. Our responses to your queries are provided below.
>
> **Limited clinical coverage and scalability concerns: the evaluation contains only to pulmonary MediTOD and small MIMIC-IV subset. This limits generalizability to other medical fields which may have different reasoning patterns and disease/symptom structures.**
>
> We thank you for highlighting the importance of clinical coverage and generalizability. We address these points below:
>
> **Expanded Evaluation on MIMIC-IV:** To address concerns regarding sample size and robustness, we significantly expanded our evaluation of the MIMIC-IV dataset. The test set now includes nearly 297 unique patient cases. Please refer to our general comment for results on this expanded test set.
>
> **Generalizability to Other Domains:** We agree that extending validation beyond the pulmonary domain is an important step; however, our current work is constrained by the MediTOD dataset, which provides annotated task-oriented dialogues for only a single medical specialty. Nonetheless, our methodology remains domain-agnostic: the BayesNet construction and LLM-based parameter estimation are designed to transfer to any medical domain supported by clinical guidelines.
>
> **In addition, the BayesNet structure was constructed via programmatic extraction followed by manual validation, where not reproducibly specified and the manual step also limits the system's applicability to new medical domains.**
>
> We clarify that the “manual validation” step consisted primarily of lightweight text cleanup rather than clinical review. Its purpose was to correct minor formatting inconsistencies such as symptoms appearing across multiple lines, embedded within paragraphs, or presented in non-standard formats. We processed all 34 diseases in under one day, using an LLM-developed tool that demonstrates the process is scalable and does not require expensive medical expertise. We will release our extraction scripts and cleanup guidelines to ensure full reproducibility.
>
> To further address scalability, we explored automating this step entirely. A preliminary test using Gemini 2.5 Flash successfully extracted 77% of symptoms directly from source webpages with a zero-shot prompt. This suggests that modern LLMs can effectively replace manual cleanup, allowing for easy extension to new medical domains without human bottlenecks.
>
> **Validation relies on comparing primary and other edges' likelihood averages, rather than benchmarking against an external ground truth using real-world epidemiological data or clinical expert consensus. The risk of the LLM encoding inaccurate or biased associations remains unanalyzed.**
>
> We agree that comparing edge types is an indirect proxy and that validation against an external ground truth is ideal. However, gold-standard probability tables for the specific MediTOD disease-symptom pairs are not publicly available.
>
> To address the concern about potential LLM bias or inaccuracy without ground truth, we conducted an external validation using NLICE (Zaid et al.), a publicly available medical Bayesian Network containing 51 diseases and 385 disease–symptom pairs with established probability tables.
>
> Using our proposed self-consistency prompting method with Gemma 2 27B, we estimated the priors and likelihoods for the NLICE variables and compared them to the ground truth distributions using **Average KL Divergence**. We benchmarked this against random guessing and a fixed probability baseline (0.5). Further, as per the suggestion of Reviewer aWD2, we check the sensitivity of the estimates to the number of samples in self-consistency and the choice of LLM.
>
> | Model | Average KL Divergence |
> | :---: | :---: |
> | Random | 0.490 |
> | Fixed 0.5 | 0.442 |
> | Gemma2 27B (50 samples) | **0.234** |
> | Gemma2 27B (10 samples) | 0.240 |
> | Qwen3 32B (50 samples) | 0.314 |
>
> The significantly lower KL divergence with LLM variants demonstrates that our self-consistency-based estimation captures meaningful probabilistic medical associations and is not merely encoding random or biased noise. Further, increasing the sampling budget improves the quality of the estimates. Finally, the choice of LLM is important as performance worsens with the Qwen3 32B model compared to Gemma2 27B.  The results indicate that our parameter estimation procedure generalizes well to established medical knowledge.
>
> *References:*
>
> * *Al-Ars, Zaid, et al. "Nlice: Synthetic medical record generation for effective primary healthcare differential diagnosis." 2023 IEEE 23rd International Conference on Bioinformatics and Bioengineering (BIBE). IEEE, 2023\.*

---

> > ### Author Response · Authors · 2025-12-03
> >
> > **The core inference simplification (Eq. 2\) is central of the method to make Bayesian inference tractable. However, patients in real-world clinical settings often have multiple concurrent diagnoses (comorbidities), which this model cannot naturally handle. Although model claims extensibility, is there any evidence to support this?**
> >
> > We acknowledge that patients in real-world clinical settings often present with multiple concurrent conditions. However, our initial model employed a single-disease assumption, positing that a patient suffers from exactly one principal diagnosis. This simplification helps keep the Bayesian inference computationally tractable by avoiding the exponential complexity ($2^N$) of the full powerset of diseases.
> >
> > Here, we propose an extension of our current Bayesian Network to handle comorbidities. We extend our Bayesian Network to model comorbidities by relaxing the single-disease assumption while avoiding the exponential computational complexity. Our approach limits the search space to single diseases and specific, clinically validated disease pairs. We used ChatGPT to generate candidate comorbidity pairs across 34 diseases in the MediTOD dataset. Following verification by a medical professional, we refined this list to 9 high-probability comorbidity pairs (e.g., pulmonary fibrosis and malignant neoplasm of lung).
> >
> > We model disease relationships using a pairwise interaction framework. Let $C$ be the set of validated disease pairs. The joint probability of a disease configuration $(D\_1, \\dots, D\_n)$ is defined as:
> >
> > $$P(D\_1, \\dots, D\_n) \= \\frac{1}{Z}e^{-E(D\_1, \\dots, D\_n)}$$
> >
> > The configuration score $E$ captures both individual and joint risks:
> >
> > $$E(D\_1, \\dots, D\_n) \= \-\\sum\_{j} h\_j D\_j \- \\sum\_{(j, k) \\in C} J\_{jk} D\_j D\_k$$
> >
> > Here, $J_{jk}$ quantifies the interaction strength between comorbid diseases, while $ h_{j} $ represents the intrinsic risk of disease $ D_{j} $. To calibrate the model, we estimate $h_j$ using the prior probabilities for each disease, adjusting for the probability mass contributed by its comorbid partners. We fixed $J_{jk} = 1$ based on validation performance, leaving further optimization for future work.
> >
> > To perform inference, we compute the posterior probability of disease $D\_j$ by summing the likelihood of it occurring in isolation and the likelihood of it occurring with a valid comorbid pair. Given the observed symptoms ($\\mathcal{S}^+, \\mathcal{S}^-$), the posterior is:
> >
> > $$P(D\_j \= 1 \\mid \\mathcal{S}^+, \\mathcal{S}^-) \= P(D\_j \= 1, \\text{others}=0 \\mid \\mathcal{S}^+, \\mathcal{S}^-) \+ \\sum\_{k : (D\_j, D\_k) \\in C} P(D\_j \= 1, D\_k \= 1, \\text{others}=0 \\mid \\mathcal{S}^+, \\mathcal{S}^-)$$
> >
> > By restricting valid configurations to single diseases and the set $C$, the state space grows linearly ($N \+ |C|$) rather than exponentially. This ensures that exact inference remains computationally tractable. We validated this extended model on the MediTOD and extended MIMIC datasets.
> >
> > | Model | MediTOD Turn Level |  | MIMIC Simulation |  |  |  |
> > | :---- | :---- | :---- | :---- | :---- | :---- | :---- |
> > |  | Medical F1 | Precision @ Infinity | MRR | Hit@1 | Hit@3 | Hit@5 |
> > | ProbMedTOD (Llama3 8B) | 0.262 | 0.442 | 0.263 | 0.152 | 0.354 | 0.485 |
> > | ProbMedTOD with Comorbidity | 0.234 | 0.430 | 0.268 | 0.173 | 0.333 | 0.458 |
> >
> > We posit that this drop in performance is due to the MediTOD and MIMIC subsets focusing mainly on a single primary diagnosis. As a result, predicting additional clinical relevant conditions can lower scores under the current setting. Future work can explore creating training and evaluation datasets with clear comorbidities.

---

> > > ### Author Response · Authors · 2025-12-03
> > >
> > > **The model sometimes asks non-informative questions (Sec. 7), is there any systematic analysis or mitigation strategy?**
> > > Our analysis of simulated patient–doctor dialogues from the MIMIC dataset reveals several patterns in which ProbMedTOD generates non-informative inquiries. The key findings are presented below. We provide the example dialogue in the next comment.
> > >
> > > * ProbMedTOD frequently conducts interviews that average approximately 52 turns (about 104 utterances). Although extensive, these interactions align with OSCE-style interviews represented in the MediTOD dataset and therefore remain consistent with the model’s training distribution.
> > > * The model follows a consistent and structured clinical interview format. It proceeds from the chief complaint to the history of present illness, then to medical history, family history, and lifestyle factors, and often includes an exhaustive review of systems. This organization reflects standard clinical practice.
> > > * As seen in lines 102–113, the model sometimes iterates through entities redundantly. This stems from a mismatch between the coarse action labels in MediTOD (e.g., single label `allergies`) and the fine-grained entities in MIMIC. The model selects the broad action *inquire allergies*, and the generator attempts to fill every slot (drug) associated with that action, resulting in repetitive clarification questions.
> > > * The model sometimes repeats questions that the patient has already answered. For example, the patient denies chest pain early in the conversation (lines 28-29), yet the doctor asks about chest pain again later (lines 100-101). Similarly, information about medication lapses is elicited more than once (lines 60-61, lines 120-121), and details about medication allergies are repeatedly requested, even after the patient explicitly states that they do not know the specifics (lines 102-113). These repeated inquiries do not contribute new information to the conversation.
> > >
> > > To mitigate repeated questions, we propose a Rejection Sampling strategy during inference. If the model generates an action identical to one generated in the past, we reject it and resample (up to 10 trials). We evaluated this using a naive implementation (due to API constraints preventing direct logit masking). Even with this simple constraint, we observed consistent improvements:
> > >
> > > | Model | MRR | Hit@1 | Hit@3 | Hit@5 |
> > > | :---: | :---: | :---: | :---: | :---: |
> > > | ProbMedTOD (Llama3 8B) | 0.263 | 0.152 | 0.354 | 0.485 |
> > > | w/ Rejection Sampling | 0.270 | 0.155 | 0.357 | 0.478 |
> > >
> > > The results demonstrate that even a naive rejection sampling strategy yields improvements in MRR, Hit@1, and Hit@3 metrics. While we observe a marginal decline in Hit@5 (from 0.485 to 0.478), this is an expected consequence of pruning high-probability tokens. These improvements suggest redundancy is, in part, a decoding problem.

---

> > > > ### Author Response · Authors · 2025-12-03
> > > >
> > > > **Example Dialogue**
> > > >
> > > > ```
> > > > 1: Patient: I've been feeling really short of breath lately, especially when I walk up the stairs.
> > > > 2: Doctor: OK, and when did this start?
> > > > 3: Patient: It started about two days ago, doctor.
> > > > 4: Doctor: Two days ago, OK, has it been getting any better, worse, or staying the same?
> > > > 5: Patient: It was getting worse, doctor. I was having more trouble breathing, especially when I walked around, and the coughing was really bothering me. I felt like there was rattling in my chest. But I do feel a bit better since I got those treatments in the ER.
> > > > 6: Doctor: OK, and have you ever had anything like this happen to you before?
> > > > 7: Patient: Yes, doctor, I have. I had a bad spell a while back, I had to go to the hospital for it. It was because of my COPD, they said.
> > > > 8: Doctor: OK, and how long ago was that?
> > > > 9: Patient: Oh, goodness, I'm not sure of the exact date, doctor. It was quite a while ago, though.
> > > > 10: Doctor: OK, and were you diagnosed with COPD at that time? Or was it after that?
> > > > 11: Patient: Oh, I think they told me I had COPD before that last hospital visit, doctor. It was the reason I had to go in.
> > > > …
> > > > 27: Patient: Yes, doctor, I do. I've been feeling wheezy, especially with this cough.
> > > > 28: Doctor: OK, and do you have any chest pain?
> > > > 29: Patient: No, doctor, I haven't had any chest pain.
> > > > 30: Doctor: OK, do you ever feel like your heart's racing?
> > > > …
> > > > 60: Doctor: OK, and what medications are you currently taking?
> > > > 61: Patient: Well, doctor, I take a little white pill for my arthritis every day, and I take something for my blood pressure. I'm also on a pill for my bones, and another one for my stomach. I take a water pill too, but I ran out of that a few days ago. I also take aspirin and something for my heart, but I stopped taking that one. Oh, and I take a pain pill when my back or hands hurt. I think that's about it.
> > > > …
> > > > 100: Doctor: OK, and do you have any chest pain?
> > > > 101: Patient: No, doctor, I haven't had any chest pain.
> > > > 102: Doctor: OK, and do you have any allergies to medications?
> > > > 103: Patient: Yes, doctor, I have allergies. I'm allergic to Penicillins, and a few other things like Darvocet-N, Talwin Nx, Demerol, and Vistaril.
> > > > 104: Doctor: OK, and do you know what you're allergic to Vistaril?
> > > > 105: Patient: No, doctor, I don't know what I'm allergic to. I just know I can't take it.
> > > > 106: Doctor: OK, and do you know what you're allergic to Demerol?
> > > > 107: Patient: No, doctor, I don't know what I'm allergic to. I just know I can't take it.
> > > > 108: Doctor: OK, and do you know what you're allergic to Talwin Nx?
> > > > 109: Patient: No, doctor, I don't know what I'm allergic to. I just know I can't take it.
> > > > 110: Doctor: OK, and do you know what you're allergic to Darvocet-N?
> > > > 111: Patient: No, doctor, I don't know what I'm allergic to. I just know I can't take it.
> > > > 112: Doctor: OK, and do you know what you're allergic to Penicillin?
> > > > 113: Patient: No, doctor, I don't know what I'm allergic to. I just know I can't take it.
> > > > 114: Doctor: OK, and are you up to date with all your immunizations?
> > > > 115: Patient: Oh, goodness, doctor, I'm not sure about my immunizations. I haven't really thought about them in a while. I get my flu shot every year, though.
> > > > 116: Doctor: OK, and when was your last pneumovax vaccine?
> > > > 117: Patient: Oh, goodness, doctor, I'm not sure about my immunizations. I haven't really thought about them in a while. I get my flu shot every year, though.
> > > > 118: Doctor: OK, and have you ever been diagnosed with congestive heart failure?
> > > > 119: Patient: No, doctor, I don't think I've ever been told I have congestive heart failure. I take that water pill, Lasix, but I'm not sure why. I ran out a few days ago.
> > > > 120: Doctor: OK, and you haven't taken it in the last few days?
> > > > 121: Patient: No, doctor, I haven't taken it in the last few days. I ran out, like I said.
> > > > ```

---

### Author Response · Authors · 2025-12-01
**General Response to All Reviewers**

We thank the reviewers for their time and valuable feedback. We have made every effort to address your queries and resolve the concerns raised during the review process.

In response to Reviewer aWD2’s suggestion regarding extending diagnoses beyond a single principal diagnosis, we utilized the Bayesian Network (BayesNet) to generate a final differential diagnosis. The revised ProbMedTOD design operates as follows: As discussed in the paper, the model constructs a diagnostic hypothesis at each turn using the BayesNet. A Policy (POL) LLM, trained on the MediTOD dataset, then predicts the next doctor action based on the dialogue context and this hypothesis. When the dialogue concludes (indicated by a salutation action from the POL model), ProbMedTOD outputs the top-5 diseases from the latest diagnostic hypothesis as the final differential.

This method marks a departure from our previous approach, which used a beam-search with a width of 5 to determine the diagnosis. This new design offers two distinct advantages: it naturally outputs multiple diagnoses and does not require retraining our models.

To measure the efficacy of this update, we ran simulations on an **expanded test set of 297 (from 84\) patient cases** from the MIMIC-IV dataset. Our revised results are reported in the table below.

| Model | MIMIC-IV |  |  |  |
| :---: | :---: | :---: | :---: | :---: |
|  | MRR | Hit@1 | Hit@3 | Hit@5 |
| Llama3 8B | 0.154 | 0.081 | 0.195 | 0.320 |
| Qwen3 8B | 0.168 | 0.084 | 0.175 | 0.374 |
| RAG (Llama3 8B) | 0.175 | 0.077 | 0.216 | 0.384 |
| ProbMedTOD (Llama3 8B) | **0.263\*** | **0.152\*** | **0.354\*** | **0.485\*** |
| ProbMedTOD (Qwen3 8B) | **0.311\*** | **0.202\*** | **0.387\*** | **0.529\*** |

\* \- indicates that the improvement of ProbMedTOD over the best baseline (RAG) is statistically significant (p < 0.05) using an Approximate Randomization Test (Carterette, et. al. 2007\) with 10,000 permutations.

As shown in the results, ProbMedTOD outperforms the baseline models across all metrics. For instance, ProbMedTOD (Llama3 8B) achieves an MRR of 0.263 and a Hit@5 score of 0.485, surpassing both the base Llama3 model and the RAG baseline. The performance improves further with the Qwen3 backbone, where ProbMedTOD achieves the highest results with an MRR of 0.311 and a Hit@5 score of 0.529. We would revise the figures in the main paper with our new findings.

References:

* Mark D. Smucker, James Allan, and Ben Carterette. 2007\. A comparison of statistical significance tests for information retrieval evaluation. In Proceedings of the sixteenth ACM conference on Conference on information and knowledge management (CIKM '07). Association for Computing Machinery, New York, NY, USA, 623–632. https://doi.org/10.1145/1321440.1321528*

---

### Meta-Review · Area_Chair_xBUS · 2025-12-28

**Summary:**

The authors propose a LLM-BayesNet hybrid model for medical diagnosis. Reviewers generally noted concerns about (1) the scalability of the approach, as the BN edges were scraped and manually filtered; (2) the validation of the constructed BN; (3) and technical limitations of the approach, such as the constraint that each patient has exactly one disease. In their rebuttal, the authors added additional experiments to address (2) and (3). They also showed that their model, while helpful on Llama3-8B and Qwen3-8B, actually makes larger models like Gemini-2.5-Flash worse, potentially because of the fixed ontology and corresponding scalability concerns in (1). Consequently, this method is not quite useful off-the-shelf right now. The main technical innovation is in the construction of the BN (with edge weights estimated by LLM).

Reviewers were overall split (4466); my guess is that the scores would have been revised slightly upward (say 5566) after rebuttal, leaving the paper borderline. I recommend that the authors revise their paper and submit to a future conference, with more emphasis on the technical contribution of the paper (i.e., BN construction and incorporation) and understanding how/why it can work with larger models.

**Reviewer Concerns:**

See main comment.

**Reviewer Scores:**

See main comment.

---

### Decision · Program_Chairs · 2026-01-26

Reject